# Mebendazole Increases Anticancer Activity of Radiotherapy in Radiotherapy-Resistant Triple-Negative Breast Cancer Cells by Enhancing Natural Killer Cell-Mediated Cytotoxicity

**DOI:** 10.3390/ijms232415493

**Published:** 2022-12-07

**Authors:** Hoon Sik Choi, Young Shin Ko, Hana Jin, Ki Mun Kang, In Bong Ha, Hojin Jeong, Jeong-hee Lee, Bae Kwon Jeong, Hye Jung Kim

**Affiliations:** 1Department of Radiation Oncology, Gyeongsang National University Changwon Hospital, Gyeongsang National University College of Medicine, Changwon 51472, Republic of Korea; 2Institute of Health Science, Gyeongsang National University, Jinju 52727, Republic of Korea; 3Biomedical Research Institute, Gyeongsang National University Hospital, Jinju 52727, Republic of Korea; 4Department of Pharmacology, Gyeongsang National University College of Medicine, Jinju 52727, Republic of Korea; 5Department of Convergence Medical Science (BK21 Plus), Gyeongsang National University College of Medicine, Jinju 52727, Republic of Korea; 6Department of Radiation Oncology, Gyeongsang National University Hospital, Gyeongsang National University College of Medicine, Jinju 52727, Republic of Korea; 7Department of Pathology, Gyeongsang National University Hospital, Gyeongsang National University College of Medicine, Jinju 52727, Republic of Korea

**Keywords:** breast neoplasms, mebendazole, radiotherapy, natural killer cell

## Abstract

Breast cancer is the most commonly diagnosed cancer worldwide and ranks first in terms of both prevalence and cancer-related mortality in women. In this study, we aimed to evaluate the anticancer effect of mebendazole (MBZ) and radiotherapy (RT) concomitant use in triple-negative breast cancer (TNBC) cells and elucidate the underlying mechanisms of action. Breast cancer mouse models and several types of breast cancer cells, including TNBC-derived RT-resistant (RT-R) MDA-MB-231 cells, were treated with MBZ and/or RT. In mice, changes in body weight, renal and liver toxicity, tumor volume, and number of lung metastases were determined. In cells, cell viability, colony formation, scratch wound healing, Matrigel invasion, and protein expression using western blotting were determined. Our findings showed that MBZ and RT combined treatment increased the anticancer effect of RT without additional toxicity. In addition, we noted that cyclin B1, PH2AX, and natural killer (NK) cell-mediated cytotoxicity increased following MBZ + RT treatment compared to unaided RT. Our results suggest that MBZ + RT have an enhanced anticancer effect in TNBC which acquires radiation resistance through blocking cell cycle progression, initiating DNA double-strand breaks, and promoting NK cell-mediated cytotoxicity.

## 1. Introduction

Breast cancer is the most commonly diagnosed cancer worldwide and ranks first in terms of both prevalence and disease-related mortality in women [1]. At the 2011 St. Gallen International Breast Cancer Conference, breast cancer was classified according to molecular subtype: luminal A [estrogen receptor (ER)+ and/or progesterone receptor (PR)+, Ki67 low, and human epidermal growth factor receptor 2 (HER2)-]; luminal B (ER+ and/or PR+, Ki-67 high and/or HER2+); HER2-positive (ER-, PR- and HER2+); triple-negative types (ER-, PR-, and HER2-) [2]. Triple-negative breast cancer (TNBC) accounts for approximately 12–17% of all breast cancers and has the lowest survival rate compared to other subtypes [3]. The poor treatment outcome in patients with TNBC is due to the lack of expression of the ER, PR, and HER2 during disease progression; thus, existing anticancer drugs targeting these receptors cannot be used.

Radiotherapy (RT), one of the main treatment methods for TNBC, causes DNA double-strand breaks, in turn causing cancer cell death. However, mechanisms that result in acquiring radioresistance, including dose limitations due to radiation tolerance of surrounding normal tissues, accelerated tumor cell repopulation, hypoxia, and enhanced repair of radiation damage can reduce the effectiveness of this therapeutic option [4]. TNBC is known to be a representative radiation-resistant tumor because of its self-renewal and regeneration characteristics, similar to cancer stem cells, based on the expression of cell surface markers CD44^+^CD24^−^ and aldehyde dehydrogenase 1 (ALDH1) [5,6]. Because of this characteristic, TNBC easily acquires resistance to existing standard treatment methods, including RT [6]. Owing to this, new treatment strategies, such as the development of novel drugs or the application of various combinations of existing therapeutic strategies, are required for the management of TNBC.

In a previous study, we selected benzimidazole derivatives as potential anticancer candidates because their anthelmintic mechanisms are associated with oncogenic pathways. Consequently, these agents displayed anticancer effects in several cell lines and animal studies without causing serious side effects [7,8]. Further, we found that mebendazole (MBZ), a benzimidazole derivative, exhibited the most effective anticancer effects in TNBC through inducing DNA damage, cell cycle arrest, and downregulating cancer stem cell markers CD44 and OCT3/4 and cancer progression-related ESM-1 protein expression [9]. Moreover, other studies that researched on cancer stem cells or cells resistant to approved treatment options reported that MBZ monotherapy has anticancer effects as a repurposed drug, and it also stimulated anti-tumor immune responses in synergy with ionizing radiation and various chemotherapeutic agents [10,11,12]. Herein, we investigated the anticancer effect and possible mechanisms of MBZ and RT coadministration in TNBC cells in vivo and in vitro.

## 2. Results

### 2.1. MBZ Shows Increased Anticancer Effect in RT-Treated Breast Cancer Mouse Models without Hepatic or Renal Toxicity

First, we investigated the combined anticancer effects of MBZ and RT in a breast cancer mouse model. A schematic diagram of the experimental schedule is shown in Figure 1A. In our previous study, MBZ hardly affected cell viability at doses of 0.01, 0.1, 0.5, 1, 2, 5, and 10 μM for 24 h in MDA-MB-231 cells and RT-R-MDA-MB-231 cells. However, when incubated for 72 h after 10 μM of MBZ administration, cell viability was slightly decreased by approximately 35–50% compared to the control [9]. Based on this result, 10 mg/kg of MBZ was tested for in vivo study and showed no hepatic damage, kidney toxicity, or body weight loss, but exhibited a significant decrease in tumor volume and lung metastasis. Therefore, in this study, we used 10 mg/kg of MBZ in the combination with RT. In addition, in the preliminary experiments, when we used a radiation dose of 20 Gy in two fractions with an irradiation interval of 2 days, mice showed significant and progressive weight loss and slowed movement after RT administration, indicating that the 20 Gy RT dose was too strong to adequately balance the positive effects of irradiation. Then, 8 Gy once/week * 3 times, 10 Gy 2 times, or 12 Gy 1 time were performed. The results showed that 12 Gy once was the most suitable for our intended experimental purpose and showed the most effective results. Therefore, we used 12 Gy once and combined it with a non-toxic dose (10 mg/kg) of MBZ. To compare the combined effects of MBZ and RT, the animals treated with PTX, an anticancer drug frequently used to treat breast cancer, were divided into seven groups as follows: (1) mice injected with 4T1 cells (4T1 group), (2) mice injected with RT-R-4T1 cells (RT-R-4T1 group), (3) RT-R-4T1 + MBZ group, (4) RT-R-4T1 + PTX group, (5) RT-R-4T1 + RT group, (6) RT-R-4T1 + RT + MBZ group, and (7) RT-R-4T1 + RT + PTX group. Treatment-related toxicity was measured by survival rate changes in the weight of mice, plasma creatinine (an indication of renal function), and plasma ALT and AST (indicators of liver function) (Figure 1B–F). No significant weight change was observed in any of the seven groups, including the RT-R-4T1 + RT + MBZ group. In addition, a significant increase in plasma creatinine, ALT, and AST was not induced by MBZ, PTX, or RT individually or in combination.

After treatment with MBZ (10 mg/kg), PTX (10 mg/kg), and RT (12 Gy), tumor volumes and numbers of lung metastases were determined to establish the anticancer effect of the therapeutic options (Figure 2). Compared to the non-treated RT-R-4T1 group, the MBZ, PTX, and RT, alone or combination with treatment groups, showed a tendency to decrease tumor volume. In particular, the RT + MBZ group showed the greatest decrease in tumor volume (Figure 2A). Twenty-eight days after the injection of cancer cells, mice were killed, and the tumor volume (Figure 2B) and the number of lung metastases (Figure 2C) were measured. Similar to the pattern observed in the changing tumor volumes, the tumor volume on day 28 showed a statistically significant decrease in the MBZ, PTX, and RT individual treatment groups compared to the non-treated RT-R-4T1 group. In addition, the RT + MBZ combined treatment group showed a statistically significant reduction in tumor volume compared with the RT unaided group. The number of lung metastases showed a statistically significant increase in the RT-R-4T1 group, inferring acquired treatment resistance compared to animals in the 4T1 group. The number of lung metastases was significantly decreased in the MBZ, PTX, and RT individual groups. Further, the RT + MBZ group showed a statistically significant reduction in the number of lung metastases compared with the RT unaided group. Therefore, these results suggest that in mice injected with RT-R-4T1 cells, the combined treatment with MBZ (10 mg/kg) and RT (12 Gy) is a safe and more efficacious regimen to reduce tumor size and lung metastasis when compared with RT alone.

### 2.2. MBZ Increases Anticancer Effect in RT-Treated TNBC in Terms of Cell Viability and Colony Formation

Cell viability and colony formation assays were performed using MCF10A, a normal epithelial cell line of the breast, and various radiation-resistant breast cancer cell lines (Figure 3). A schematic of the experimental schedule is shown in Figure 3A. In the cell viability assay by CCK-8 assay, MBZ was administered at 0.1 or 0.5 μM concentrations, and RT was irradiated at 5 or 10 Gy. In MCF10A cells (Figure 3B), MBZ or RT alone induced a slight decrease in cell viability but showed an absolute viability > 80% compared with non-treated MCF10A. The coadministration of MBZ and RT also induced a slight decrease in cell viability; however, the absolute cell viability was maintained at approximately 80% compared with non-treated MCF10A. After the exact treatment as aforementioned, cell viability was measured in three radiation-resistant cell lines (RT-R-MDA-MB-231, RT-R-MCF7, and RT-R-T47D) (Figure 3C). In all three cell lines, MBZ monotherapy did not have a significant effect on cell viability. Contrarily, in all three cell lines, RT alone induced a statistically significant decrease in cell viability compared with non-treated each cancer cell. In addition, in RT-R-MDA-MB-231 and RT-R-MCF cells, a positive dose-dependent relationship was observed. In the case of MBZ and RT coadministration, MBZ (0.5 μM) induced a statistically significant additional reduction in cell viability in all three cell lines and at all radiation doses compared with RT alone treated group. When 0.1 μM MBZ was administered, the additional reduction in cell viability was statistically significant only when 10 Gy was irradiated in RT-R-MCF7 cells, while both 5 Gy and 10 Gy doses showed significant outcomes in RT-R-T47D cells. This enhanced effect was more pronounced in RT-R-MDA-MB-231 cells derived from TNBC than in the other two cell lines. Therefore, the combined effect of MBZ and RT was further investigated in RT-R-MDA-MB-231 cells derived from TNBC based on a colony formation assay, at a reduced dose of 3 Gy. The assay results showed that MBZ or RT individual treatments decreased the colony-forming ability compared to the control group, and this effect was significantly amplified when RT was combined with MBZ (Figure 3D). These results suggest that the combined regimen of MBZ and RT is relatively non-toxic to normal breast tissue cells, but more effectively reduces cell viability and the colony-forming ability of RT-R-TNBC cells.

### 2.3. Combined Therapy of MBZ and RT More Reduces Cell Migration and Invasion of RT-R-MDA-MB-231 Cells Compared to RT Alone

To investigate the effect of MBZ and RT individual treatments, as well as their combination on migration, a scratch migration assay was performed on RT-R-MDA-MB-231 cells. RT-R-MDA-MB-231 cells were scratched with a sterile 200 μL pipette tip, treated with MBZ (0.5 μM), and irradiated or not with a 3 Gy dose of X-rays. After 24 h, MBZ and RT individual treatments had significantly reduced the number of cells that migrated into the scratched areas compared with non-treated cells, and this effect was further increased significantly by combining MBZ with RT (Figure 4A,B). Similarly, the combined treatment with MBZ and RT reduced the invasive property of RT-R-MB-MB-231 cells across the Matrigel-EC coated transmembrane compared to RT alone (Figure 4C,D). These results suggest that MBZ enhances the effect of RT on limiting cell migration and invasion in the TNBC-derived cell line, RT-R-MDA-MB-231.

### 2.4. Combination of MBZ and RT Effectively Enhances the Cyclin B1 Expression and pH2AX Level of RT-R-MDA-MB-231 Cells

To observe cell cycle arrest and apoptosis, levels of cyclin B1 and cleaved caspase-3 were measured. In RT-R-MDA-MB-231 cells, MBZ (0.5 and 1 μM) monotherapy significantly increased the expression of cyclin B1, while RT similarly increased the expression of cyclin B1 compared with the control group. Interestingly, the combined treatment with MBZ and RT significantly increased the expression of cyclin B1 compared to the individual agents (Figure 5A). MBZ (0.5, 1 μM) and RT alone significantly induced cleaved caspase-3, however, no additive or synergistic effects were observed from the combination regimen (Figure 5B). To observe DNA damage, the levels of pH2AX, a DNA double-strand break marker, were measured. The level of pH2AX was significantly increased in cells individually treated with MBZ (0.5, 1 μM) and RT. In addition, MBZ prominently enhanced RT-mediated pH2AX induction at concentrations of 0.5 and 1 μM (Figure 5C). Overall, MBZ + RT co-treatment did not have an additive effect on increasing cleaved caspase-3 but further increased cyclin B1 and pH2AX.

### 2.5. Combination of MBZ Enhances Natural Killer (NK) Cell-Mediated Cytotoxicity in RT Treatment

In the previous study, RT-R-MDA-MB-231 cells decreased the cytotoxicity of NK cells compared to MDA-MB-231 through down-regulating ligands for NK cell activation [13]. Therefore, in this study, we determined the effect of MBZ on NK-cell mediated cytotoxicity in RT-treated RT-R-MDA-MB-231 cells. NK cell-mediated target cell killing was measured using an LDH cytotoxicity assay. In the assay, RT-R-MDA-MB-231 cells were used as target cells and NK cells were used as effector cells at different target: effector cell ratios, as shown in Figure 6A. At a target: effector cell ratio of 1:10, MBZ (0.5, 1 μM) treatment significantly increased NK cell-mediated cytotoxicity (Figure 6B). Nonetheless, at the same cell ratio, RT (3 Gy) did not increase NK cell-mediated cytotoxicity; however, co-treatment with MBZ (0.5 μM) significantly increased NK cell cytotoxicity (Figure 6C). Next, the released levels of perforin and granzyme B, which are proteins related to NK cell-mediated cytotoxicity, were measured in the culture media (Figure 7). When RT-R-MDA-MB-231 cells and NK cells were co-incubated, RT alone did not increase the secretion of perforin and granzyme B. However, MBZ + RT co-treatment significantly increased perforin and granzyme B release from NK cells compared to the individual therapeutic options.

## 3. Discussion

TNBC is highly invasive, has high metastatic potential, and is prone to relapse, resulting in a poor prognosis with a mortality rate of 40% within 5 years [14,15]. Furthermore, TNBC does not respond to hormone- or trastuzumab-based therapies because of the lack of target receptors such as ER, PR, and HER2 in the tumor-related environments. Owing to this, a combination of surgery, chemotherapy, and RT appears to be the only available modality [16]. Because novel treatment options are cardinal to managing the disease, we investigated the anticancer effect of combined MBZ and RT treatment.

MBZ, a benzimidazole derivative, induces various responses in vivo, and has anthelmintic effects [17]. In the field of oncology, research is being conducted to supplement existing treatments for various cancers by harnessing the anticancer effect of anthelmintics [8]. In a previous study, we found that MBZ alone had anticancer effects in RT-resistant TNBC by inducing apoptosis and cell cycle arrest, suppressing the expression of cancer stem cell markers (CD44 and OCT3/4), and inhibiting cancer progression-related ESM-1 proteins [9]. Herein, we found that the coadministration of MBZ and RT increased the anticancer effect in RT-resistant TNBC compared to MBZ or RT individual regimens.

We first determined the toxicity of MBZ (10 mg/kg) + RT (12 Gy) by measuring changes in body weight, plasma creatinine, plasma ALT, and plasma AST in mice injected with RT-R-4T1 breast cancer cells and by examining the viability of MCF10A cells, a normal breast epithelial cell line. The results from these experiments demonstrated that co-treatment with MBZ (10 mg/kg) and RT (12 Gy) had little or no toxicity. Based on our findings and the proven safety profile of MBZ, the MBZ+RT combination regimen can be considered a safe treatment option that has no toxicity to normal tissues.

Following combination treatment with MBZ and RT at the determined safe doses, the anticancer effect of the regimen was determined in animal models and using cell line experiments. The main measurement point was whether the anticancer effect of the combined treatment was greater than that of RT. From the results of tumor volume, number of lung metastases, tumor cell viability, colony formation, tumor cell migration, and invasion, the enhanced anticancer effect of the combined regimen was demonstrated in RT-R-TNBC cells.

In this study, the enhanced anticancer effect of the combination regimen was associated with increased cyclin B1 expression. In a previous study, MBZ induced cell cycle arrest in the G2/M phase by increasing cyclin B1 expression and inhibiting tubulin polymerization [9]. From this result, it can be inferred that MBZ + RT can promote cell cycle arrest in the G2/M phase through an increase in cyclin B1. In addition, in this study, PH2AX was significantly increased in the MBZ + RT combined treatment group, confirming the increased effect on DNA double-strand breaks.

Le et al. conducted a study titled “Mebendazole potentiates radiation therapy in triple-negative breast cancer” [18]. They reported that breast cancer-initiating cells (BCIC) are hierarchically constituting cells of breast cancer and are resistant to chemotherapy or RT, and TNBC is relatively rich in BCIC, making it readily resistant to conventional treatment. They also reported that non-tumorigenic breast cancer cells that survive exposure to ionizing radiation can be converted to BCIC by the activation of a dedifferentiation program. They performed high-throughput screening to identify compounds that inhibit RT-induced dedifferentiation in TNBC and confirmed MBZ as a potential agent. In their study, they argued that MBZ could induce cell cycle arrest, DNA double-strand breaks, and apoptosis, as well as act as a radiation sensitizer by reducing BCIC. This conclusion supports the enhanced anticancer effect of MBZ, as observed herein.

The normal immune system generally contributes to the recognition and removal of foreign pathogens including tumors [19]. In particular, NK cells, a subset of innate lymphoid cells, are the most efficient effector cells among cytolytic lymphocytes, showing natural cytotoxicity against primary tumor cells and suppressing metastasis through inhibition of cancer cell proliferation, migration, and colonization to distant organs [20]. However, interestingly, the immune cells in the tumor microenvironment (TME) exhibit a functional change and contribute to growth and metastasis rather than the removal of tumor cells. The NK cell function is also inhibited in the TME and provides conditions that are favorable for tumor progression. According to our previous study [13], RT-R-TNBC cells which exhibit more aggressive properties in terms of tumor growth, invasion, and metastasis suppressed NK cell cytotoxicity by regulating ligands for NK cell activation. RT-R-MDA-MB-231 decreased the expression levels of MICA/B, the ligands for the activating NKG2D receptor, but increased the expression of HLA-E, the ligand for the inhibitory NKG2A receptor. Moreover, NK-92MI cells cocultured with RT-R-MDA-MB-231 cells decreased the levels of secreted perforin and granzyme B more than NK-92MI cells cultured with MDA-MB-231 cells. Therefore, in this study, we further analyzed whether the combination of MBZ and RT affects the cytotoxic function of NK cells, with one of the white blood cells responsible for the innate immune system. Although NK cells mainly act to dissolve cells infected with viruses or bacteria, they can directly inhibit the development, proliferation, and metastasis of cancer cells and effectively eliminate cancer stem cells, which are important for recurrence [21]. In this study, MBZ + RT significantly increased NK cell-mediated cytotoxicity. In addition, perforin and granzyme B, which are proteins secreted from activated NK cells that play a major role in destroying target cells, were significantly increased in the MBZ + RT group. In contrast, no further increase in cleaved caspase-3 expression was observed. Thus, it could be speculated that NK cells are partly involved in the enhanced anticancer effect of MBZ + RT, and among associated actions, the granule-mediated mechanism of NK cells may be involved. In addition, we postulate the involvement of a caspase-independent pathway involving the mitochondria for cell destruction. Although studies have shown that MBZ or RT individually activate NK cells as an immunomodulator [22,23], this study is the first to show an additional enhanced effect from coadministration.

## 4. Materials and Methods

### 4.1. Materials

MBZ (Figure 8) and paclitaxel (PTX) were purchased from Sigma-Aldrich (St. Louis, MO, USA). MBZ was resuspended in 0.25% sodium carboxymethyl cellulose (CMC) solution. RPMI 1640 medium, fetal bovine serum (FBS), and antibiotics (penicillin/streptomycin) were purchased from Cytiva (Marlborough, MA, USA). Anti-cyclin B1 antibody was purchased from Abcam (Cambridge, UK). Anti-pH2AX and cleaved caspase-3 antibody were purchased from Cell Signaling Technology (Beverly, MA, USA). The BD Matrigel basement membrane matrix was supplied by BD Biosciences (San Diego, CA, USA). The Cyto Tox 96^®^ Non-Radioactive Cytotoxicity Assay Kit was obtained from Promega (Madison, WI, USA). The human perforin and granzyme B ELISA Kits were obtained from MyBioSource (San Diego, CA, USA). The enhanced chemiluminescence (ECL) western blotting detection reagent was obtained from Bio-Rad (Hercules, CA, USA). All other chemicals were purchased from Sigma-Aldrich (St. Louis, MO, USA).

### 4.2. Animal Experiments

Female athymic nude mice (6-weeks-old) were purchased from Orient Bio (Gyeonggi-do, Republic of Korea). Mice were housed under the following constant ambient conditions: 22–26 °C, 40–60% relative humidity, 12 h light/dark cycles, and free access to sterilized food and water. The experimental protocol was approved by the Institutional Animal Care and Use Committee of the Gyeongsang National University (approval number: GNU-200603-N0030). The mice were injected into the right thigh with 4T1 or RT-resistant (RT-R)-4T1 cells (5 × 10^4^ cells/100 μL). Seven days after the injection, body weight and tumor volume were measured three times per week. Fourteen days after injection, the mice were divided into seven groups (n = 7 per group): (1) mice injected with 4T1 cells (4T1 group), (2) mice injected with RT-R-4T1 cells (RT-R-4T1 group), (3) RT-R-4T1 + MBZ group, (4) RT-R-4T1 + PTX group, (5) RT-R-4T1 + RT group, (6) RT-R-4T1 + RT + MBZ group, and (7) RT-R-4T1 + RT + PTX group. For RT, the mice were anesthetized by intramuscular injection of Zoletil and irradiated with a single 12 Gy dose of X-rays to the right thigh tumor using a 6 MV photon beam from a linear accelerator (21EX, Varian, Palo Alto, CA, USA). On the day of irradiation, MBZ (10 mg/kg) or PTX (10 mg/kg) was administered by oral gavage and intraperitoneal injection, respectively, followed by daily treatment for two consecutive weeks on a “5 days on and 2 days off” schedule. The mice were killed on the 28^th^ day after the initial injection, tumor volumes and lung metastases were measured, and blood was collected. Blood plasma was separated by centrifuging at 3000× *g* for 15 min. Plasma alanine aminotransferase (ALT) and aspartate aminotransferase (AST) levels were measured using assay kits from the IVD Lab (Uiwang, Republic of Korea) and a spectrophotometer (Shimadzu UV-1800 spectrophotometer, Tokyo, Japan). Plasma creatinine levels were measured directly using the colorimetric Jaffe method.

### 4.3. Cell Cultures

Human breast cancer cell lines, MDA-MB-231, MCF7, and T47D, were obtained from the Korea Cell Line Bank (Seoul, Republic of Korea). The non-malignant breast epithelial cell line, MCF10A, mouse breast cancer cell line, 4T1, human umbilical endothelial cell line, EA.hy926, and human NK cell line, NK-92MI, were obtained from American Type Culture Collection (ATCC; Manassas, VA, USA). RT-R-MDA-MB-231, RT-R-MCF7, RT-R-T47D, and RT-R-4T1 cells were generated by repetitively applying a 2 Gy dose of X-rays until a final dose of 50 Gy was achieved. All cancer cell lines were cultured in RPMI 1640 supplemented with 10% FBS, 100 IU/mL penicillin, and 10 μg/mL streptomycin, and incubated at 37 °C in a humidified atmosphere containing 5% CO_2_. NK-92MI cells were cultured in alpha-MEM containing 2 mM L-glutamine, 1.5 g/L sodium bicarbonate (GIBCO; Thermo Fisher Scientific, Waltham, MA, USA) supplemented with 0.2 mM inositol (Sigma-Aldrich, St. Louis, WA, USA), 0.1 mM 2-mercaptoethanol (Sigma-Aldrich, St. Louis, WA, USA), 0.02 mM folic acid (Sigma-Aldrich, St. Louis, WA, USA), 12.5% horse serum (GIBCO), and 12.5% FBS without ribonucleosides and deoxyribonucleosides at 37 °C in a humidified atmosphere containing 5% CO_2_.

### 4.4. Cell Viability Assay

Cells were seeded at a density of 1 × 10^4^ cells/well in 24-well plates, treated with MBZ (0.1 or 0.5 μM) for 24 h, and then irradiated with different doses of X-rays (5 Gy, 10 Gy). After 24 h, cells were treated with 10 μL/well CCK-8 reagent (Dongin Biotech, Seoul, Republic of Korea) and incubated for 30 min at 37 °C in the dark. The optical density of each well was measured at 450 nm using a microplate reader (Molecular Devices VersaMax, Sunnyvale, CA, USA).

### 4.5. Colony Formation Assay

RT-R-MDA-MB-231 cells were cultured in 6-well plates. The cells were treated with MBZ (0.1 or 0.5 μM) for 24 h at 37 °C and then irradiated with a 3 Gy dose of X-rays. After 24 h, the cells were trypsinized, centrifuged at 1000 rpm for 2 min, and resuspended in RPMI 1640 medium. The cells were seeded in 6-well plates (1 × 10^3^ cells/well) and the culture medium was replaced with a completely new medium every 2–3 days. After 10 days, the medium was discarded, and the cells were washed with phosphate-buffered saline (PBS). The colonies were fixed in 100% methanol for 10 min at 25 °C, stained with 0.1% Giemsa staining solution for 30 min at the same temperature, and visible colonies were counted.

### 4.6. Scratch Migration Assay

RT-R-MDA-MB-231 cells were cultured in 6-well plates and scratched with a sterile 200 μL pipette tip. The cells were treated with MBZ (0.5 μM) for 24 h at 37 °C and then irradiated with a 3 Gy dose of X-rays. After 24 h, the cells were washed with PBS and images were taken using an Olympus photomicroscope at 0 and 48 h after scratching. The number of cells that migrated to the scratched area was determined.

### 4.7. Matrigel Invasion Assay

RT-R-MDA-MB-231 cells were cultured in 6-well plates and the cells were treated with MBZ (0.5 μM) at 37 °C for 24 h, followed by irradiation with a 3 Gy dose of X-rays. After 24 h, the cells were trypsinized, centrifuged at 1000 rpm for 2 min, and resuspended in RPMI 1640 medium. The upper chambers of the inserts were coated with 100 μL of Matrigel (1 mg/mL), and endothelial cells (Ecs) (2 × 10^5^ cells) were added to the Matrigel-coated inserts. Resuspended RT-R-MDA-MB-231 cells (2 × 10^5^ cells/insert) were added to the upper chambers in serum-free media, and the lower chambers were filled with 500 µL of RPMI-1640 containing 10% FBS. The cells were grown at 37 °C for 16 h in a culture incubator. To remove non-invasive cells, the upper surface of the insert membranes was scrubbed with a wet swab. Subsequently, the cells on the lower insert membranes were stained with DAPI (Sigma-Aldrich), and the invading cells were counted under a fluorescence microscope (Eclipse Ti-U, Nikon, Tokyo, Japan).

### 4.8. Western Blot Analysis

RT-R-MDA-MB-231 cells were treated with MBZ (0.1, 0.5, and 1 μM) for 1 h and then irradiated with a 3 Gy dose of X-rays. After 24 h, the cells were harvested and lysed in RIPA buffer containing 50 mM Tris-HCl (pH 7.5), 150 mM NaCl, 1% NP-40, 0.1% SDS, 0.5% sodium deoxycholate, and protease inhibitors. The samples were centrifuged at 13,000 rpm for 20 min at 4 °C and the supernatants were collected. Protein concentration was determined using the Bradford method. Equal amounts of protein were subjected to between 8–10% sodium dodecyl sulfate-polyacrylamide gel electrophoresis (SDS-PAGE) and transferred onto Hybond-P + polyvinylidene difluoride membranes (Amersham, Buckinghamshire, UK). The membranes were blocked with 5% non-fat milk in Tris-buffered saline containing 0.05% Tween-20 (TBS-T) for 1 h at 25 °C and then incubated with the following primary antibodies: anti-cyclin B1 (ab32053; 1:1000), anti-cleaved caspase-3 (9661S; 1:1000), and anti-p-H2AX (2577S; 1:1000). The bound antibodies were detected using horseradish peroxidase-conjugated secondary antibodies and an ECL (Bio-Rad, Hercules, CA, USA) western blotting detection reagent. The relative protein levels were normalized to that of β-actin (MA5-15739, 1:5000, Thermo Fisher Scientific, Waltham, MA, USA), which was used as a loading control.

### 4.9. Cytotoxicity Assay

NK cell cytotoxicity against cancer cells was assessed using the Cyto Tox 96^®^ Non-Radioactive Cytotoxicity Assay Kit (Promega). RT-R-MDA-MB-231 cells (5 × 10^3^ cells/60 μL) were seeded in a round-bottom 96-well plate and NK-92MI cells were added. After co-incubation for 4 h at 37 °C, the culture medium was centrifuged at 1000 rpm for 1 min, and the supernatant was obtained. To determine the effect of MBZ on NK cell-mediated cytotoxicity against cancer cells, RT-R-MDA-MB-231 cells were treated with MBZ (0.1, 0.5, and 1 μM) for 24 h, and then the cells (5 × 10^3^ cells/60 μL) were seeded in a round-bottom 96-well plate. NK-92MI cells were added at a 1:10 ratio (cancer cell: NK cell) in the 96-well plate containing RT-R-MDA-MB-231 cells. For MBZ + RT co-treatment, RT-R-MDA-MB-231 cells were treated with MBZ (0.5 μM) for 1 h and then irradiated with a 3 Gy dose of X-rays. After 24 h, the cells (5 × 10^3^ cells/60 μL) were seeded in a round-bottom 96-well plate and NK-92MI cells were added at a 1:10 ratio (cancer cell: NK cell) in the 96-well plate containing RT-R-MDA-MB-231 cells. The levels of lactate dehydrogenase (LDH) released into the supernatant were determined as described in the manufacturer’s protocol. Cytotoxicity was quantified as follows:$$Percent\;of\;cytotoxicity=\frac{Experimental\;LDH\;Release\;\left(OD490\right)}{Maximum\;LDH\;Release\;\left(OD490\right)}\times 100$$

### 4.10. Measurement of Extracellular Perforin and Granzyme B Levels

The extracellular levels of perforin and granzyme B were measured using Human perforin and granzyme ELISA Kits (MyBioSource, San Diego, CA, USA), respectively. RT-R-MDA-MB-231 cells were pretreated with MBZ (0.5 μM) for 1 h before being irradiated with a 3 Gy dose of X-rays. After 24 h, the cells (5 × 10^3^ cells/60 μL) were seeded in a round-bottom 96-well plate, and NK-92MI cells (cancer cell: NK cell ratio = 1:10) were added. After co-incubation for 4 h at 37 °C, supernatants were obtained from the culture medium by centrifugation at 1000 rpm for 1 min. Perforin and granzyme B levels in cell supernatants were measured according to the manufacturer’s instructions.

### 4.11. Statistical Analysis

All data were statistically analyzed using GraphPad Prism 7 software (GraphPad Software, San Diego, CA, USA). One-way ANOVA followed by Tukey’s multiple comparison test was performed to compare differences among groups. Data are presented as mean ± standard deviation (SD).

## 5. Conclusions

In conclusion, this study underscores the following: (1) the need for additional therapeutic strategies for TNBC, (2) the safety of a MBZ + RT regimen, (3) the enhanced anticancer effect of MBZ + RT, and (4) that various targets are involved in the overall mechanism of action of this combined therapy. Taken together, our results suggest that MBZ + RT co-treatment has an enhanced anticancer effect in TNBC, which acquires radiation resistance by blocking cell cycle progression, producing DNA double-strand breaks, and promoting NK cell-mediated cytotoxicity.

## Figures and Tables

**Figure 1 ijms-23-15493-f001:**
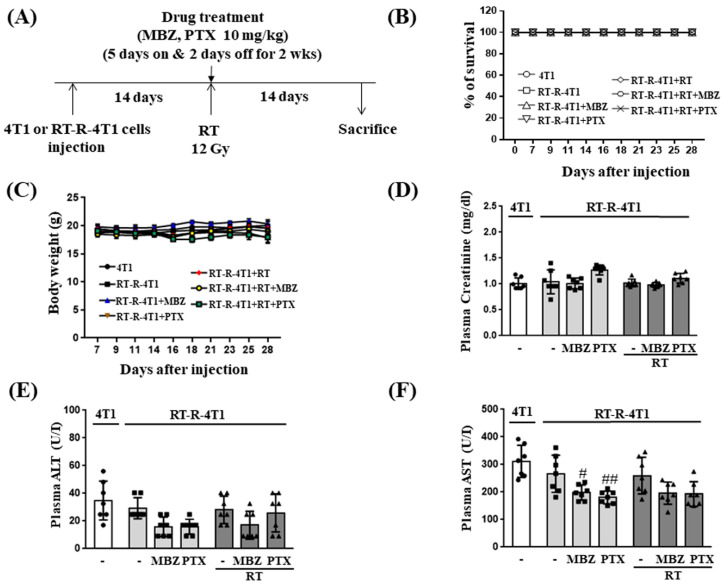
The combination of mebendazole (MBZ) and radiotherapy (RT) has no serious toxicity on RT-resistant-4T1 (RT-R-4T1) tumor allografts in athymic nude mice. (**A**) Experimental scheme for 4T1 and RT-R-4T1 allograft construction and drug treatment in athymic nude mice. 4T1 cells or RT-R-4T1 cells (5 × 10^4^ cells/100 μL) were injected into the right thighs and the mice were divided into 7 groups (n = 7 per group) on day 14 after injection; (1) mice injected with 4T1 cells (4T1 group), (2) mice injected with RT-R-4T1 cells (RT-R- 4T1 group), (3) RT-R-4T1 group + MBZ, (4) RT-R-4T1 group + paclitaxel (PTX), (5) RT-R-4T1 group + RT, (6) RT-R-4T1 group + RT + MBZ, (7) RT-R-4T1 group + RT + PTX. For RT, the mice were irradiated with a single dose of 12 Gy to the right thigh. MBZ (10 mg/kg) and PTX (10 mg/kg) were administered by oral gavage and intraperitoneal injection, respectively, on the same day before irradiation and thereafter daily for two consecutive weeks on a “5 days on and 2 days off” schedule. (**B**,**C**) Body weights and survival rate were measured three times a week for four weeks from day seven after the injection of cancer cells. The mice were killed on the 28th day after injection, and levels of (**D**) plasma creatinine, (**E**) alanine aminotransferase (ALT), and (**F**) aspartate aminotransferase (AST) were measured. The bar graph in white is for 4T1 cells, light gray is for RT-R-4T1 cell treated with MBZ or PTX alone, and dark gray is for RT-R-4T1 cell with RT combined treatment. The values are presented as the mean ± SD from five independent determinations (^#^
*p* < 0.05, ^##^
*p* < 0.01 compared with the control of RT-R-4T1).

**Figure 2 ijms-23-15493-f002:**
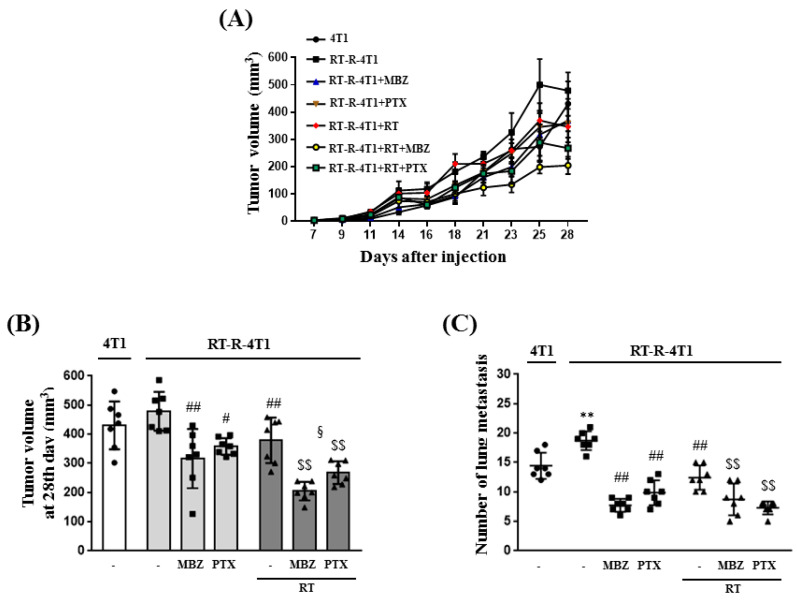
The combination of mebendazole (MBZ) and radiotherapy (RT) increases the anticancer effect of RT on RT-resistant-4T1 (RT-R-4T1) tumor allografts in athymic nude mice. (**A**) Tumor volumes were measured three times a week for four weeks from day seven after the injection of cancer cells. The mice were killed on the 28th day after injection, and tumor volumes (**B**) and number of lung metastases (**C**) were measured. The white bar graph is the tumor volume of 4T1 cells, light gray is the tumor volume of the MBZ or paclitaxel (PTX) alone treatment group of RT-R-4T1 cells, and dark gray is the tumor volume of the RT combined treatment group of RT-R-4T1 cells. The values are presented as the mean ± SD from five independent determinations (** *p* < 0.01 compared with 4T1; ^#^
*p* < 0.05, ^##^
*p* < 0.01 compared with the control of RT-R-4T1; ^$$^
*p* < 0.01 compared with the control of RT; ^§^
*p* < 0.05 compared with RT-R-4T1 + MBZ).

**Figure 3 ijms-23-15493-f003:**
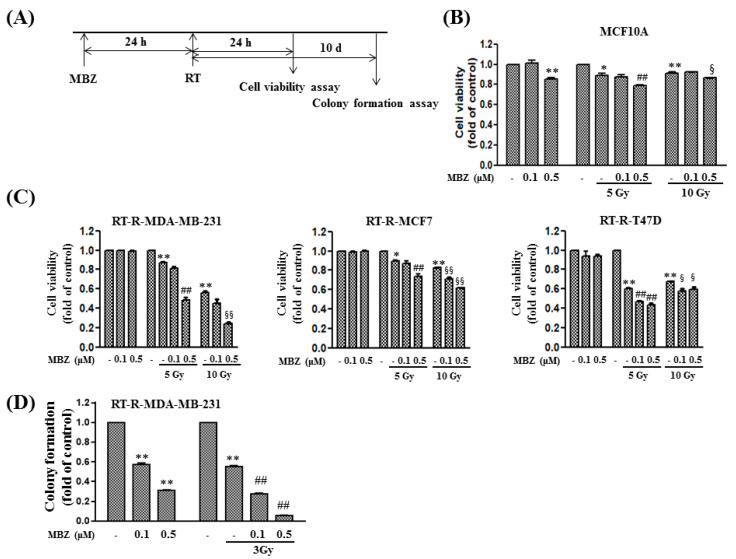
The combination of mebendazole (MBZ) and radiotherapy (RT) effectively reduces cell viability of triple negative breast cancer-derived RT-resistant-MDA-MB-231 (RT-R-MDA-MB-231) cells without causing serious cytotoxicity in normal epithelial MCF-10A cells. (**A**) Experimental scheme. (**B**) Normal breast epithelial cells, MCF10A, and (**C**) RT-resistant (RT-R) breast cancer cells, RT-R-MDA-MB-231, RT-R-MCF-7, and RT-R-T47D were treated with MBZ at the indicated concentrations (0.1, 0.5 μM) for 24 h and then irradiated with different does of X-rays (5 Gy, 10 Gy). After 24 h, cell viability was measured using the CCK-8 assay kit as described in the Materials and Methods. The data are represented as the mean ± standard deviation (SD) (* *p* < 0.05, ** *p* < 0.01 compared with the control; ^##^
*p* < 0.01 compared with the control of RT 5 Gy; ^§^
*p* < 0.05, ^§§^
*p* < 0.01 compared with the control of RT 10 Gy). (**D**) RT-R-MDA-MB-231 cells were treated with MBZ at the indicated concentrations (0.1, 0.5 μM) for 24 h and then irradiated with a 3 Gy dose of X-rays. After 24 h, the cells were seeded in 6-well plates (1 × 10^3^ cells/well). The culture medium was replaced with complete medium every 2–3 days. After 10 days, the colonies were fixed, stained with 0.1% Giemsa staining solution, and visible colonies were counted. The data are presented as the mean ± SD (** *p* < 0.01 compared with the control; ^##^
*p* < 0.01 compared with the control of RT).

**Figure 4 ijms-23-15493-f004:**
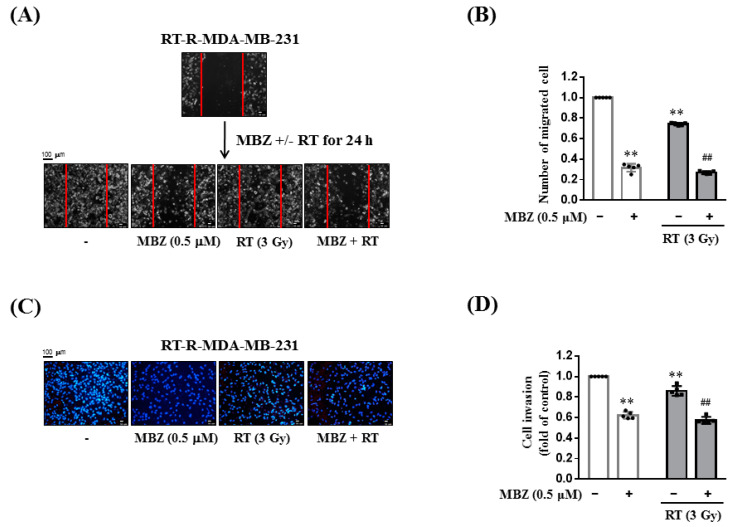
The combination of mebendazole (MBZ) and radiotherapy (RT) reduces scratch migration and cell invasion more effectively than RT monotherapy. (**A**,**B**) RT-resistant (RT-R)-MDA-MB-231 cells were cultured in 6-well plates and scratched with a sterile 200 μL pipette tip. The cells were treated with MBZ (0.5 μM) for 24 h and then irradiated with a 3 Gy dose of X-rays. After 24 h, cell images (**A**) were taken using an Olympus photomicroscope at 0 and 48 h after scratching and (**B**) numbers of cells that migrated into the scratch area were quantified. (**C**,**D**) RT-R-MDA-MB-231 cells were treated with MBZ (0.5 μM) for 24 h and then irradiated with a 3Gy dose of X-rays. After 24 h, the cells were added to EC-Matrigel-coated insert wells and incubated for 16 h. The cells that invaded across the membrane were stained with (**C**) DAPI and (**D**) the invading cells were counted under a fluorescence microscope. The white bar graph represents the MBZ alone treatment group, and the gray bar graph represents the MBZ and radiotherapy combination group. The values are presented as the mean ± SD from five independent determinations (** *p* < 0.01 compared with the control; ^##^
*p* < 0.01 compared with the control of RT).

**Figure 5 ijms-23-15493-f005:**
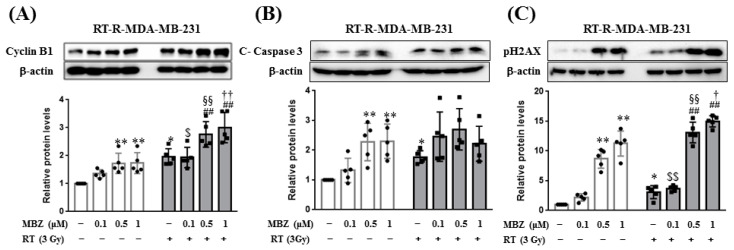
The combination of mebendazole (MBZ) and radiotherapy (RT) significantly induces cyclin B1 expression and the levels of the DNA damage marker, pH2AX, compared to RT. RT-resistant (RT-R)-MDA-MB-231 cells were treated with MBZ (0.1, 0.5, and 1 μM) for 1 h and then irradiated with a 3 Gy dose of X-rays. After 24 h, the protein expression of (**A**) cyclin B1, (**B**) cleaved caspase-3, and (**C**) pH2AX were determined in cell lysates by western blotting as described in the Materials and Methods. The white bar graph represents the MBZ alone treatment group, and the gray bar graph represents the MBZ and radiotherapy combination group. The values are presented as the mean ± SD from five independent determinations (* *p* < 0.05, ** *p* < 0.01 compared with the control; ^##^
*p* < 0.01 compared with the control of RT 3 Gy; ^$^
*p* < 0.05, ^$$^
*p* < 0.01 compared with MBZ (0.1 μM); ^§§^
*p* < 0.01 compared with MBZ (0.5 μM); ^†^
*p* < 0.05, ^††^
*p* < 0.01 compared with MBZ (1 μM)).

**Figure 6 ijms-23-15493-f006:**
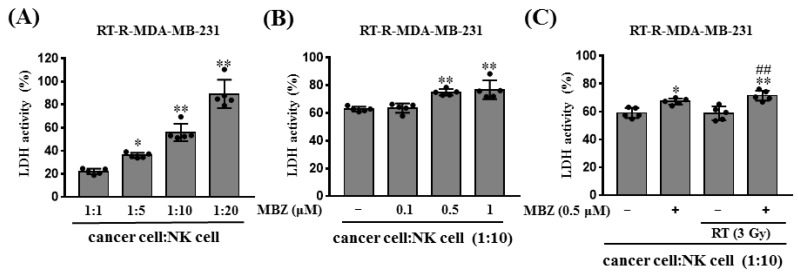
The combination of mebendazole (MBZ) and radiotherapy (RT) significantly increases NK cell-mediated cytotoxicity against RT resistant (RT-R)-MDA-MB-231 cells. (**A**) RT-R-MDA-MB-231 cells (5 × 10^3^ cells/60 μL) were seeded in round-bottom 96-well plates, and NK-92MI cells were added at the indicated ratios (1:1, 1:5, 1:10, and 1:20). After co-incubation for 4 h, LDH was measured in cell supernatants. (**B**) RT-R-MDA-MB-231 cells were treated with MBZ (0.1, 0.5, and 1 μM) for 24 h, and the cells were collected and seeded in a round-bottom 96-well plate (5 × 10^3^ cells/60 μL/well). Next, NK-92MI cells (1:10 ratio) were added to the 96-well plate, co-incubated for 4 h, and LDH was measured in the cell supernatant as described in the Methods. (**C**) RT-R-MDA-MB-231 cells were treated with MBZ (0.5 μM) for 1 h and then irradiated with a 3Gy dose of X-rays. After 24 h, the cells (5 × 10^3^ cells/60 μL) were seeded in a round-bottom 96-well plate, and NK-92MI cells (1:10 ratio) were added. After co-incubation for 4 h, LDH was measured in the cell supernatant as described above. The values are presented as the mean ± SD (* *p* < 0.05, ** *p* < 0.01 compared with the control; ^##^
*p* < 0.01 compared with the control of RT 3 Gy).

**Figure 7 ijms-23-15493-f007:**
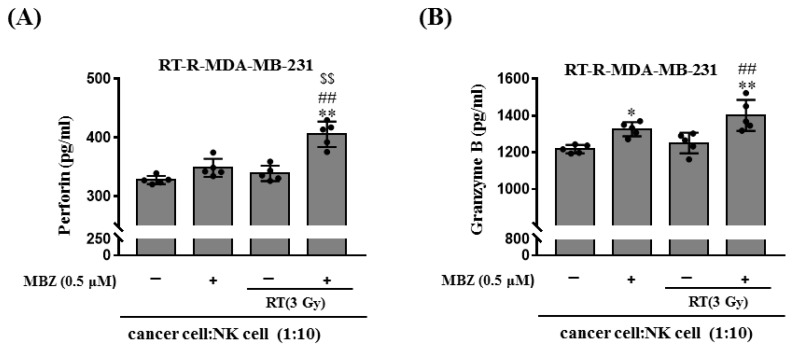
The combination of mebendazole (MBZ) and radiotherapy (RT) more efficiently induces perforin and granzyme B production by NK cells compared to RT. RT-resistant (RT-R)-MDA-MB-231 cells were treated with MBZ (0.5 μM) for 1 h and then irradiated with a 3 Gy dose of X-rays. After 24 h, the cells were seeded in a round-bottom 96-well plate, and NK-92MI cells (1:10 ratio) were added. After co-incubation for 4 h, (**A**) perforin and (**B**) granzyme B were measured in cell supernatants as described in the manufacturer’s protocol. The values are presented as the mean ± SD (* *p* < 0.05, ** *p* < 0.01 compared with the control; ^##^
*p* < 0.01 compared with the control of RT; ^$$^
*p* < 0.01 compared with MBZ).

**Figure 8 ijms-23-15493-f008:**
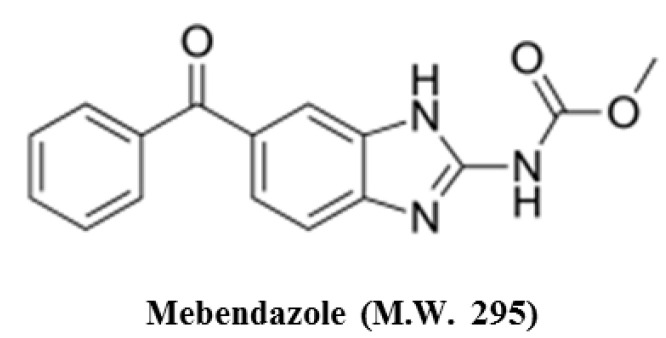
Structure of mebendazole.

## Data Availability

The data presented in this study are available upon request.

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
