# Peer review of "Mebendazole Increases Anticancer Activity of Radiotherapy in Radiotherapy-Resistant Triple-Negative Breast Cancer Cells by Enhancing Natural Killer Cell-Mediated Cytotoxicity"

_ijms, 2022, doi:10.3390/ijms232415493_

Round 1

Reviewer 1 Report

The manuscript titled "Mebendazole increases anticancer activity of radiotherapy in radiotherapy-resistant triple-negative breast cancer cells by enhancing natural killer cell-mediated cytotoxicity" investigates the anticancer effect and possible mechanisms of MBZ and RT co-administration in triple-negative breast cancer cells in vivo and in vitro. The work is interesting and has some useful data, but major changes are required.

1) The authors should explain why they used an irradiation dose of 12 Gy in vivo. The survival curve of the mice should be present. Also what is the LD50/30 for the mice? Is it 12 Gy? What methods did they use to determine this dose?

2) Why did the authors use these precise doses of radiation, i.e. 5, 3 or 10 Gy, in vitro? What methods did they use to determine these doses? Are they LD50 or D0 for the cell lines?

3) The used concentrations of MBZ are based on IC50? What methods did the authors use to determine these concentrations?

4) The individual charts and pictures are really small. They are difficult to read. The authors should modify them accordingly.

5) The authors should revise the superscripts throughout the article (103 → 103).

6) Figures 4A and 4C are too small to see any fluorescent signal.

7) The authors should clearly describe the differences between the control groups.

Author Response

Reviewer #1

The manuscript titled "Mebendazole increases anticancer activity of radiotherapy in radiotherapy-resistant triple-negative breast cancer cells by enhancing natural killer cell-mediated cytotoxicity" investigates the anticancer effect and possible mechanisms of MBZ and RT co-administration in triple-negative breast cancer cells in vivo and in vitro. The work is interesting and has some useful data, but major changes are required.

1) The authors should explain why they used an irradiation dose of 12 Gy in vivo. The survival curve of the mice should be present. Also what is the LD50/30 for the mice? Is it 12 Gy? What methods did they use to determine this dose?

Response:

Thank you for your valuable question.

As we explained in the "Introduction", patients with triple-negative breast cancer (TNBC) show poor treatment outcomes for the use of anticancer drugs targeting these receptors due to the lack of expression of the ER, PR, and HER2. Thus, existing anticancer drugs targeting these receptors cannot be used. Therefore, radiotherapy (RT) is one of the main treatment methods for TNBC. However, TNBC easily acquires resistance to RT and recurs and resulting in treatment failure. Therefore, the development of novel drugs or the application of various combinations of existing therapeutic strategies is required for the management of TNBC. For this purpose, we sought to investigate the increased anticancer effect of RT in combination with mebendazole without serious side effects on RT-resistant TNBCs.

We conducted several preliminary experiments to determine the appropriate radiation dose. In the first attempt, a dose of 20 Gy/2 fractions was used with an irradiation interval of 2 days, referring to the radiation dose of other similar studies. As a result, the mice showed significant and gradual weight loss and slowed movements, so we judged that the 20 Gy dose was too strong for the purpose of the experiment and adjusted it. Then, the same experiment was performed on the 24 Gy/3 fractions, 10 Gy/2 fractions, or 12 Gy/1 fraction, respectively. The results showed that 12 Gy/1 fraction was the most suitable for our intended experimental purpose and showed the most effective results. Therefore, we used 12 Gy/1 fraction and combined it with a non-toxic dose of mebendazole. All doses of RT used in our experiments had no effect on the survival of mouse. Therefore, we couldn’t get the LD30/50.

2) Why did the authors use these precise doses of radiation, i.e. 5, 3 or 10 Gy, in vitro? What methods did they use to determine these doses? Are they LD50 or D0 for the cell lines?

Response:

As we explained above, we tested several doses of RT to get a safe dose of RT in vitro by CCK-8 assay (lines 144, 173).

3) The used concentrations of MBZ are based on IC50? What methods did the authors use to determine these concentrations?

Response:

In the previous study [Molecules. 2021 Aug 24;26(17):5118], we determined the effects of mebendazole at concentrations of 0.01, 0.1, 0.5, 1, 2, 5, and 10 μM on the viability of the normal breast epithelial cell line MCF10A, TNBC cell line MDA-MB-231 and RT-R-MDA-MB-231 by the CCK assay. Regarding normal breast epithelial cells, MCF10A, none of the drugs affected the cell viability at doses 0.01, 0.1, 0.5, and 1 μM for 24–72 h. Regarding TNBC cells, for MDA-MB-231 and RT-R-MDA-MB-231 cells, the MBZ hardly affected cell viability at doses of 0.01, 0.1, 0.5, 1, 2, 5, and 10 μM for 24 h. However, when incubated for 72 h after 10 μM of MBZ administration, cell viability was slightly decreased, by approximately 35–50% compared to the control. Because we used a safe dose of mebendazole with a survival rate of over 80%, 0.1 and 0.5 μM, the IC50 was not calculated, as we explained above.

4) The individual charts and pictures are really small. They are difficult to read. The authors should modify them accordingly.

Response:

The size of the inserted chart and figure has been corrected.

5) The authors should revise the superscripts throughout the article (103 → 103).

Response:

Thank you for your comment.

We corrected that scale into 103 or 104 in red color (please see lines 123, 177, Figure 6 legends, lines 365, 375, 392, 393, 419, 424, 428, and 438).

6) Figures 4A and 4C are too small to see any fluorescent signal.

Response:

The size of the inserted chart and figure has been corrected.

7) The authors should clearly describe the differences between the control groups.

Response:

Thank you for your comments.

As you indicated, we edited some sentences to make more clearly the meaning.

In Figure 2, we compared the anticancer effect of the combined treatment with MBZ and RT on tumor size and lung metastasis compared with RT alone (lines 109, 112, 118-120).

In Figure 3, we compared the effect of MBZ or RT alone on cell viability with non-treated cells, and then compared the effect of MBZ and RT coadministration with RT alone treated (lines 146, 149, 153, 157).

In Figure 4, the effect of MBZ and RT individual treatment on cell migration and invasion was compared with the non-treated cell, and the effect of the combined treatment with MBZ and RT was compared to RT alone (lines 188-193).

Reviewer 2 Report

1.     In section 2.2, the cell viability assay was based on the allograft-derived cells or the primary cell lines. If the assay was based on the allografts, how to ensure the RT effect was still active when the isolated cells were cultured? The same question is also for the co-administration of RT and MZT circumstance. 

2.     In section 2.2, the authors concluded that the normal breast cell was not sensitive to RT and MTZ. The mechanism of RT was causing DNA damage. Why the mechanism did not work on the normal cell?

3.     In section 2.5, when the RT-R-MDA-MB-231 and NK were co-cultured, did the NK was also exposed to RT? 

4.     The title “Mebendazole increases anticancer activity of radiotherapy in radiotherapy-resistant triple-negative breast cancer cells by enhancing natural killer cell-mediated cytotoxicity ”meant that the anticancer effect was from NK cells. Whereas, all the results(except section 2.5) indicated that RT and MTZ could work on tumor cell directly. How to explain the relevant results? 

Author Response

Reviewer #2

1) In section 2.2, the cell viability assay was based on the allograft-derived cells or the primary cell lines. If the assay was based on the allografts, how to ensure the RT effect was still active when the isolated cells were cultured? The same question is also for the co-administration of RT and MBZ circumstance.

Response:

In fact, we didn’t isolate cells from irradiated mice. In section 2.2, we used radio-resistant human breast cancer cell lines (RT-R-MCF-6, RT-R-MDA-MB-231, and RT-R-T47D) and commercial normal breast epithelial cells. To generate radio-resistant breast cancer cells, we applied repetitive small doses of X-ray irradiation (2 Gy) to each cell line until a final dose of 50 Gy was reached, which is a commonly used clinical regimen for the radiotherapy of the breast cancer patient. In the previous study (Ko et al., Oncology Reports, 2018, 40:3752-2762), we confirmed that established RT-R-breast cancer cells exhibited resistance to radiation compared to their parental breast cancer cells.

2) In section 2.2, the authors concluded that the normal breast cell was not sensitive to RT and MBZ. The mechanism of RT was causing DNA damage. Why the mechanism did not work on the normal cell?

Response:

RT works because the radiation destroys the cancer cell's ability to reproduce and the body naturally eliminates these cells. Radiation affects cancer cells by damaging their DNA, so cancer cells can no longer divide and multiply. Radiation can also affect normal cells but is more effective at killing actively dividing cells such as cancer cells. The reason cancer cells are more susceptible to radiation than normal cells is that cancer cells divide faster than normal cells and they cannot repair damage as effectively as normal cells.

We selected a dose of radiation that did not affect the cell viability of normal cells for 24 h. On the other hand, this dose of radiation induced DNA damage in cancer cells, which are more vulnerable than normal cells.

3) In section 2.5, when the RT-R-MDA-MB-231 and NK were co-cultured, did the NK was also exposed to RT?

Response:

As we described in Figure 6C legend, we treated RT-R-MDA-MB-231 cells with mebendazole (0.5 μM) for 1 h and then irradiated with a 3Gy dose of X-rays in the presence of mebendazole. After 24 hr, RT-R-MDA-MB-231 cells were harvested and seeded in a round-bottom 96-well plate, and NK-92MI cells (non-irradiated) were added.

4) The title “Mebendazole increases anticancer activity of radiotherapy in radiotherapy-resistant triple-negative breast cancer cells by enhancing natural killer cell-mediated cytotoxicity” meant that the anticancer effect was from NK cells. Whereas, all the results (except section 2.5) indicated that RT and MBZ could work on tumor cell directly. How to explain the relevant results?

Response:

In this study, we found that MBZ + RT have an enhanced anticancer effect in TNBC which acquires radiation resistance through blocking tumor cell cycle progression, initiating DNA double-strand breaks in the tumor cell. Furthermore, we found that MBZ promoted NK cell-mediated cytotoxicity to RT-R-TNBC.

As you are concerned, we didn’t show the direct connection of NK cell on anticancer effects mediated by MBZ and RT. Therefore, to avoid some confusion, we changed the sentence line 293 to “We further analyzed whether the combination of MBZ and RT affects the cytotoxic function of NK cells~.” In addition, we changed some words ‘can’ with ‘could” and inserted “partly” in line 302.

However, we would like to emphasize this point that we newly found the effect of NK cell on MBZ and RT-mediated anticancer effect, so we proposed the tile as follows; “Mebendazole increases anticancer activity of radiotherapy in radiotherapy-resistant triple-negative breast cancer cells by enhancing natural killer cell-mediated cytotoxicity.”

Round 2

Reviewer 1 Report

The authors answered all questions.

The authors mention that doses of RT used in their experiments had no effect on the survival of the mouse. However, 12 Gy is a really high dose for some of the experiments in mice, the authors should support their statement with survival charts.

Also, the authors used 10 mg/kg of MBZ and PTX. Did the authors perform an MTD (maximum tolerated dose) for the tested substances in mice? If not, why did the authors choose a concentration of 10 mg/kg? Based on which results?

The authors should again revise the superscripts throughout the article (104 → 104).

Author Response

Response to Reviewers

The authors answered all questions.

Point 1. The authors mention that doses of RT used in their experiments had no effect on the survival of the mouse. However, 12 Gy is a really high dose for some of the experiments in mice, the authors should support their statement with survival charts.

Response 1:

Thank you for your comment.

As you pointed out, a radiation dose of 12 Gy is a dangerous dose that can kill a human or a mouse when exposed to the whole body. However, as in our experiment, it is a relatively safe dose for local irradiation. Although it is difficult to directly compare, a dose of 12 Gy/1 fraction is frequently used in radiosurgery for localized cancer patients.

According to your suggestion, we changed Figure 1 with the new Figure 1 including the survival rate as Figure 1B. In addition, to help the readers’ understanding, we included the following explanation, “In addition, in the preliminary experiments, when we used a radiation dose of 20 Gy in two fractions with an irradiation interval of 2 days, mice showed significant and progressive weight loss and slowed movement after RT administration, indicating that the 20 Gy RT dose was too strong to adequately balance the positive effects of irradiation. Then, 8 Gy once/week * 3 times, 10 Gy 2 times, or 12 Gy 1 time were performed. The results showed that 12 Gy once was the most suitable for our intended experimental purpose and showed the most effective results (data not shown). Therefore, we used 12 Gy once and combined it with a non-toxic dose (10 mg/kg) of MBZ.” Please see lines 96-104.

Point 2. Also, the authors used 10 mg/kg of MBZ and PTX. Did the authors perform an MTD (maximum tolerated dose) for the tested substances in mice? If not, why did the authors choose a concentration of 10 mg/kg? Based on which results?

Response 2:

In our previous study (Choi et al., Molecules, 2021), we determined the effect of MBZ on the viability of the normal breast epithelial cell line MCF10A and TNBC cell line MDA-MB-231 and RT-R-MDA-MB-231 by the CCK assay in a dose-dependent manner. MBZ didn’t affect the cell viability of the MCF-10A at doses of 0.01, 0.1, 0,5, and 1 mM for 24–72 h. A high dose (10 mM) of MBZ decreased the cell viability of MCF10A by approximately 40–50% compared to the control at 72 h. In the case of MDA-MB-231 and RT-R-MDA-MB-231 cells, MBZ hardly affected cell viability at doses of 0.01, 0.1, 0.5, 1, 2, 5, and 10 mM for 24 h. However, when incubated for 72 h after 10 mM of MBZ administration, cell viability was slightly decreased, by approximately 35–50% compared to the control. Based on these results, we chose 10 mg/kg of MBZ for the in vivo study. 10 mg/kg of MBZ has no hepatic damage, kidney toxicity, or body weight loss, but showed a significant decrease in tumor volume and lung metastasis. Therefore, in this study, we used 10 mg/kg of MBZ in the combination with RT and also used 10 mg/kg of PTX for comparison. Please see Figure 2 and Figure 4 which were presented by Choi et al. (Molecules, 2021) for your reference.

To help the readers understand well, we included this explanation in the revised manuscript as follows, “In our previous study, MBZ hardly affected cell viability at doses of 0.01, 0.1, 0.5, 1, 2, 5, and 10 mM for 24 h in MDA-MB-231 cells and RT-R-MDA-MB-231 cells. However, when incubated for 72 h after 10 mM of MBZ administration, cell viability was slightly decreased, by approximately 35–50% compared to the control [9]. Based on this result, 10 mg/kg of MBZ was tested for in vivo study and showed no hepatic damage, kidney toxicity, or body weight loss, but exhibited a significant decrease in tumor volume and lung metastasis. Therefore, in this study, we used 10 mg/kg of MBZ in the combination with RT.” Please see lines 90-96.

Point 3. The authors should again revise the superscripts throughout the article (104 → 104).

Response 3: We changed 104 to 104. (Lines 137, 365, and 399)

Reviewer 2 Report

Dear authors,

The manuscript presented a valuable clue to treating TNBC of radiotherapy resistance. The only question is why NK cells were discussed in the study. All the results acquired were independent of NK cells except for which was explained in the Section 2.5. The authors should give convincing evidence to show the relevance of NK cells to the study.

Author Response

Response to Reviewer Comments Point 1:

Dear authors, The manuscript presented a valuable clue to treating TNBC of radiotherapy resistance. The only question is why NK cells were discussed in the study. All the results acquired were independent of NK cells except for which was explained in the Section 2.5. The authors should give convincing evidence to show the relevance of NK cells to the study.

Response 1:

Thank you for your valuable question. Radiation derives specific molecular changes sensed by the immune system, making tumor cells more susceptible to immune attack. The impacts of radiation on immune cells, such as T cells, macrophages, dendritic cells, and NK cells, have been extensively studied. NK cell is an immune cell that plays a critical role in innate immune system activation against abnormal cells. Different from events required for T cell activation, NK cell activation is controlled by the interaction of various NK receptors with target cells, independent of antigen processing and presentation. NK cells have drawn significant attention in the field of cancer immunotherapy because of their relatively simple steps for activation and their ability to potentially circumvent the shortcomings of traditional immune checkpoint blockade by utilizing the innate immune system rather than the adaptive immune system. Therefore, in this study, we observed NK cell-mediated cytotoxicity caused by MBZ + RT in radioresistant TNBC. To clarify the relationship between this study and NK cells, the following contents have been added to the “Discussion” section.

Lines 293-295

Recently, it has been reported that radiation acts as an immune modulator in addition to its direct cell-killing role [18]. The impacts of radiation on immune cells, such as T cells, macrophages, dendritic cells, and NK cells, have been extensively studied [19].

References

18. Ruckert M, Deloch L, Fietkau R, Frey B, Hecht M, Gaipl US. Immune modulatory effects of radiotherapy as basis for well-reasoned radioimmunotherapies. Strahlenther Onkol. 2018;194(6):509-1

19. Ruckert M, Flohr AS, Hecht M, Gaipl US. Radiotherapy and the immune system: More than just immune suppression. Stem Cells. 2021;39(9):1155-65.

Round 3

Reviewer 2 Report

Dear authors,

According to "Mebendazole increases anticancer activity of radiotherapy in radiotherapy-resistant triple-negative breast cancer cells by enhancing natural killer cell-mediated cytotoxicity" , ps provide evidence to the following themes.

1. In Section 2.1, 2.2, 2.3 and 2.4 all the explained effects have nothing to do with NK, could you please provide some clues in each section to show the necessity to include NK in your work?

2. The sole Section 2.5 is sufficient to support the title. 

Author Response

Response to Reviewer

Comments

Dear authors,

According to "Mebendazole increases anticancer activity of radiotherapy in radiotherapy-resistant triple-negative breast cancer cells by enhancing natural killer cell-mediated cytotoxicity", ps provide evidence to the following themes.

Point 1. In Section 2.1, 2.2, 2.3 and 2.4 all the explained effects have nothing to do with NK, could you please provide some clues in each section to show the necessity to include NK in your work?

Response 1:

Thank you for your continued interest. Our study did not analyze the NK cells due to any of the results shown in sections 2.1, 2.2, 2.3, and 2.4. Each result section is a compartmentalized collection of our observations. In section 2.1, we showed that MBZ+RT increased anticancer effects while being safe in a mouse model. In section 2.2, we showed that MBZ + RT, combined treatment, did not affect normal epithelial cell viability, but inhibited cell viability or colony formation in radiation-resistant breast cancer cell lines. In section 2.3, we showed that MBZ + RT also inhibited cancer cell migration and invasion. In section 2.4, we showed that MBZ + RT blocks tumor cell cycle progression and induces DNA double-strand breaks. We wondered if there were any new substances that MBZ + RT could target to increase the anticancer effect as above. We searched the literature on what targets are related to the mechanisms of increasing the anticancer effect of MBZ + RT, and we analyzed several markers including CD44, Oct, HMGB1, and NK cell as candidates. As a result, we found a significant difference in NK cell-mediated cytotoxicity and described it in our paper.

Point 2. The sole Section 2.5 is sufficient to support the title.

Response 2:

Our results demonstrated that MBZ + RT had an enhanced anticancer effect in TNBC, which acquired radiation resistance by blocking tumor cell cycle progression, initiating DNA double-strand breaks in tumor cells, and promoting NK cell-mediated cytotoxicity. However, we would like to emphasize the point that we newly found the effect of NK cells on MBZ + RT-mediated anticancer effect, so we suggested the title as follows; “Mebendazole increases anticancer activity of radiotherapy in radiotherapy-resistant triple-negative breast cancer cells by enhancing natural killer cell-mediated cytotoxicity.”
